# Suicide Risk in Bipolar Disorder: A Brief Review

**DOI:** 10.3390/medicina55080403

**Published:** 2019-07-24

**Authors:** Peter Dome, Zoltan Rihmer, Xenia Gonda

**Affiliations:** 1Department of Psychiatry and Psychotherapy, Semmelweis University, Faculty of Medicine, 1125 Budapest, Hungary; 2National Institute of Psychiatry and Addictions, Laboratory for Suicide Research and Prevention, 1135 Budapest, Hungary; 3MTA-SE Neuropsychopharmacology, Neurochemistry Research Group, Hungarian Academy of Sciences, 1089 Budapest, Hungary; 4NAP-2-SE New Antidepressant Target Research Group, Hungarian Brain Research Program, Semmelweis University, 1089 Budapest, Hungary

**Keywords:** bipolar disorder, mood disorders, suicide, suicidal, mortality

## Abstract

Bipolar disorders (BDs) are prevalent mental health illnesses that affect about 1–5% of the total population, have a chronic course and are associated with a markedly elevated premature mortality. One of the contributors for the decreased life expectancy in BD is suicide. Accordingly, the rate of suicide among BD patients is approximately 10–30 times higher than the corresponding rate in the general population. Extant research found that up to 20% of (mostly untreated) BD subjects end their life by suicide, and 20–60% of them attempt suicide at least one in their lifetime. In our paper we briefly recapitulate the current knowledge on the epidemiological aspects of suicide in BD as well as factors associated with suicidal risk in BD. Furthermore, we also discuss concisely the possible means of suicide prevention in BD.

## 1. Introduction

With a lifetime prevalence of 1.3–5.0%, type-I and -II bipolar disorders (BD-I; BD-II) are among the most common psychiatric ailments [1,2]. Patients with BD have poor life expectancies as these patients have a decreased lifespan of about 9–17 years compared with the general population. Furthermore, some studies from different countries (e.g., Denmark and UK) suggest that this mortality gap has become larger over the last decades. Although the largest number of excess death cases in BD may be attributed to natural (e.g., due to cardiovascular diseases or diabetes) and not unnatural causes, suicide is also quite prevalent in the population of subjects with BD [1,2,3,4,5].

At a global scale, approximately 800,000 suicide deaths occur every year (which corresponds to a global suicide rate of 11.4/100,000/year); thus, suicide may be considered a major public health issue [6,7]. Although the great majority (≈90%) of suicide cases occur among subjects with major mental—typically mood–disorders, the majority of patients with mood disorders never become involved in suicidal behaviour. Accordingly, in addition to major mood disorders, other risk factors (including special clinical features of the mental illness as well as some demographic, personality and familial factors) should contribute to suicidality, which therefore should be deemed as a multicausal phenomenon [2,8,9,10]. Hereinafter, we provide a concise summary of our current knowledge about suicidality in BD based on a review of current literature (mainly review papers, book chapters, meta-analyses, treatment guidelines of international societies, etc.).

## 2. Epidemiology of Suicidal Behaviour in Bipolar Disorder

Suicidal behaviour is quite frequent among subjects with BD, as up to 4–19% of them ultimately end their life by suicide, while 20–60% of them attempt suicide at least once in their lifetime [2]. In BD, the risk of suicide death is up to 10–30 times higher than that of the general population [2,5,8,10,11,12]. The estimated annual suicide rate in patients with BD is about 200–400 / 100,000 [8]. BD-associated cases account for about 3–14% of all suicide deaths [13].

It is important to mention that the ratio of suicide attempts to suicide deaths (i.e., the lethality index) is much lower for patients with BD than for the members of the general population (one study, for example, reported that rate as 35:1 and 3:1 for the general population and for BD patients, respectively) [2,8,9]. A possible explanation for this phenomenon may be that BD subjects usually employ more lethal suicide methods compared with members of the general population [2,8,9]. Nevertheless, attempts-to-suicide ratios lower than in the general population are not specific for BD, as it is also observable for instance among patients with schizophrenia or major depressive disorder (MDD) [2,14]. Unsurprisingly, suicidal ideation is also far more frequent in patients with BD (43% past-year prevalence) than in the general population (9.2% life-time prevalence) [7,15].

Though it is indisputable that mood disorders are associated with markedly elevated levels of suicidality, it is hard to pick out from the results of various studies whether there are *relevant* differences in the risk of suicidal behaviour between different kinds of mood disorders. Accordingly, higher, similar or lower levels of suicidality in BD patients compared to MDD patients have also been reported [9,10,16]. In a similar fashion, based on the published information it is hard to disentangle whether any BD subtype (BD-I or BD-II) is associated with a higher level of suicidality than the other [2,8,11,16,17,18,19].

It is known that a relatively high proportion (8–55%) of patients with MDD has a history of subthreshold hypomanic symptoms. This so called subthreshold bipolar subgroup of MDD patients differs from MDD patients without subthreshold hypomanic manifestations in several ways. For instance, a wide array of studies demonstrated that subthreshold bipolarity is associated with increased levels of suicidality [20,21,22,23].

## 3. Risk Factors of Suicide in Bipolar Disorder

Several approaches exist to classify risk factors for suicide in BD. One of the most common systems divides risk factors into proximal and distal ones, where proximal (or precipitating) factors are close to suicidal behaviour in time whereas distal factors are rather considered as traits or predispositions and, accordingly, they are enduring [10,24]. Other classifications assign suicide risk factors to conceptual categories (e.g., risk factors associated with genetic or sociodemographic components or illness characteristics or life events) [8,25,26]. Based on different conceptual backgrounds complex models were conceived for the description of the whole process of suicide (e.g., the diathesis-stress model, the bipolar suicidality model, the interpersonal theory of suicide, the three-step theory model or the recently elaborated “neurocognitive model of suicide in the context of bipolar disorders”) [10].

In the current paper–without the ambition to be exhaustive–we list and briefly discuss the most relevant risk and protective factors of suicide in BD. In regard to *clinical history*, *previous suicide attempt(s)* is considered as one of the most powerful single predictors of future attempts and suicide death. The *period soon after hospital discharge* may be characterized by extremely high levels of suicidality. This finding draws attention to the importance of avoiding premature discharges and inappropriate follow-ups. In addition, risk of suicide is increased during the *period immediately after hospital admission*. *Frequent and/or great number of prior hospitalizations* are also associated with heightened risk of suicidal self-harming behaviour. *Early age at onset* is also associated with suicidality in BD. The *early years after the diagnosis* represent a high-risk period for suicide. *Comorbidity* with other psychiatric, addictive or severe somatic disorders also increase the risk of all forms of suicidal behaviour. *Rapid-cycling course* and *predominant depressive polarity* during the prior course are also associated with higher risks of self-destructive behaviour. One of the most important determinants of suicidal behavior in BD is the *type/polarity of the current mood episode/state*: pure major depressive episodes and mixed states carry the highest risk, while suicidal behaviour is rarely present in (euphoric) mania, hypomania and during euthymic periods. However, some recent results indicated that there is no elevated risk of suicidal behaviour during mixed state over the risk attributable to its depressed component. Furthermore, these studies suggest that the majority of suicide risk elevation related to having previous mixed states is not an aftermath of the mixed state itself, but can rather be attributed to a depression-predominant course of the disorder. *Longer duration of untreated illness* (i.e., long time lag from the beginning of the affective symptoms until treatment initiation) is also associated with higher hazards of suicidal behaviour. Regarding *sociodemographic factors*, male *gender* is a risk factor for lethal suicides, while, according to some results, female gender is a risk factor for attempts. These gender differences are similar–but weaker–to those observable in the general population; accordingly, in this otherwise high-risk population gender seems not to be a significant predictor for suicidal behaviour). Suicidality is also more frequent among those bipolar subjects who are *divorced*, *unmarried* or *single-parents* or *living in social isolation*. *Age* is a further important sociodemographic factor: BD subjects under 35 years of age and above 75 years of age are at higher risk for engaging in suicide-related behaviours. *Occupational problems and unemployment* also contribute to elevated levels of suicidality. *Adversities in personal history and acute stressors*, such as experiencing sexual or physical abuse and parental loss in childhood or bereavement, breaking the law/criminal conviction and financial disasters are important precipitants of suicidality as well. Some *personality attributes,* for instance impulsive/aggressive traits, hopelessness and pessimism also increase the risk of suicide. Certain types of *affective temperaments* (first and foremost cyclothymic) have also been demonstrated to be associated with more frequent suicidal behaviour in BD. *Family history* of suicide acts and/or major mood disorders are also strong risk factors for suicide in subjects with BD. Some results also suggest that *living in geographical locations where there are large differences in solar insolation between winter and summer* (i.e., near the poles) may be associated with increased risks of attempted suicide in patients with BD-I [2,7,8,10,11,12,15,17,19,25,26,27,28,29,30,31,32,33,34].

## 4. Protective Factors of Suicide in Bipolar Disorder

In contrast to the above discussed several risk factors for suicide in BD, only a few protective factors have been identified so far [2]. For instance *good family and social support*, *parenthood* and *the use of adaptive coping strategies* seem to have some protective effects. Furthermore, a strong *perceived meaning of life and hyperthymic affective temperament* are also a protective factors [2,10,24,29]. The possible protective role of religiosity has emerged but results are somewhat inconclusive [2,26,35,36,37]. Last but not least, it is important to note that treatment (and even more so a good response to treatment) is protective against suicide in BD (see also the section “Suicide prevention in bipolar disorder”). In consonance with the fact that treatment may decrease heightened suicidality, it is not surprising that the majority of suicide victims are *untreated* affective disorder patients [8,9,10,11,13,38,39].

## 5. Suicide Prevention in Bipolar Disorder

From a *pharmacological perspective*, *lithium* seems to possess the greatest suicide-preventive potential in patients with BD. Intriguingly, the suicide protective effect of lithium is not confined to bipolar patients as it has also been demonstrated among patients with MDD (it is not surprising since, as we have discussed it previously, a considerable proportion of “unipolar” MDD patients have subthreshold bipolar features) [5,8,15,40,41,42]. Overall, compared to placebo, lithium appears to decrease the risk of suicide by more than 60% in mood disorders [8,40,42]. Some results suggest that lithium is protective against suicide, albeit in a decreased manner, even in those BD patients who are moderate/poor responders to the phase-prophylactic effect of it. This finding may suggest that in the case of lithium non-response in a patient who is at high risk for suicide, instead of switching lithium to another mood stabilizer, the clinician should retain lithium (even in a lower dose) and combine it with another mood stabilizer [1,41].

A solid suicide-protective effect related to the administration of *anticonvulsant-type mood stabilizers* (e.g., valproic acid, carbamazepine, lamotrigine) to BD patients has not been proven so far. On the other hand, the concern of the FDA about the potential for an increased risk of suicidality associated with anticonvulsants seems not to be applicable to patients with BD (i.e., in this population the use of these agents is not associated with increased levels of suicidality). According to our current knowledge, in regard to suicide prevention lithium is superior than these agents [2,8,15,41,43,44].

The role of *antidepressants* (ADs) in suicide prevention in individuals with BD seems to be negligible, and, in fact, concerns have been raised that administration of ADs may increase suicidality in BD. It is remarkable that findings are also inconsistent regarding the ability of ADs to prevent suicides in patients with MDD. AD monotherapy should be avoided in BD [2,8,15,41].

Considering their increasing use in BD for instance as maintenance treatment, it is justifiable to ask whether (atypical) *antipsychotics* have any beneficial effects on suicidal behaviour in BD. Unfortunately, there are no high-quality data to answer this question at present, so further studies should elucidate whether treatment with antipsychotics has any benefits in this respect [2,8,15,41].

*Ketamin* as a possible antidepressant agent has mainly been tested in patients with MDD and only a few studies have been conducted among patients with bipolar depression. According to the results of these small proof-of-concept investigations, ketamin shows similar antidepressive efficacy in bipolar as in unipolar depression. In line with its possible efficacy, ketamin is recommended by the clinical guideline of International College of Neuropsychopharmacology (CINP) for the treatment of bipolar depression, but only as a fourth-line agent and in combination with a mood stabilizer. Similarly, until now, the antisuicidal activity of ketamine was assessed mainly in MDD patients and only a small number of investigations have been conducted in BD patients. These have mainly positive outcomes, but further studies are needed to reveal whether ketamin has a similar antisuicidal effect in BD than in MDD [45,46,47,48,49,50,51].

It is well-known that *electroconvulsive therapy* (ECT) shows a similar efficacy in the treatment of depressive episodes in MDD and BPD (and some studies even found it more effective against bipolar than unipolar depression). In line with its antidepressive effects, ECT is also considered as an effective antisuicidal treatment modality, and it has been recently demonstrated that it is superior in this regard to psychopharmacons both in unipolar and bipolar depression (and its antisuicidal efficacy is comparable to the efficacy of psychopharmacons in bipolar mixed states and mania) [2,8,41,52,53].

Unfortunately, only a small number of studies have investigated up to now the efficacy of specific (e.g., dialectical behavior therapy, cognitive-behavioural therapy, interpersonal and social rhythm therapy) or unspecific (e.g., psychoeducation) *psychosocial interventions* against suicide among BD patients. Nonetheless, results of the few existing studies are promising [2,8,54,55,56,57,58].

## 6. Summary and Clinical Implications

BD is a relatively common psychiatric disorder that is associated with increased mortality due to both natural and unnatural causes. Accordingly, the risk of suicide is highly elevated in this patient population. Because of this, a thorough assessment of suicide risk should take place at all clinical visits. This clinical assessment should include, *inter alia*, the comprehensive examination of the mental state, and the inquiry about the existence and features of current suicidal intents (e.g., duration and intensity), the methods intended to be used, the access to means (e.g., weapons) as well as the compliance to prescribed medications. In addition, it is essential to gain information about previous suicidality. Whenever possible, hetero-anamnestic data should be gathered as well. The management of suicidal behaviour in patients with BD represents a clinical challenge. Appropriate long-term treatment of the disorder seems to be associated with the reduction of suicidality. Furthermore, in acutely suicidal patients the removal of access to obvious means for suicide is essential and, in severe cases, hospitalization may be justifiable as well. Prevention strategies should include the provision of psychoeducation (for example, via information leaflets and/or by the members of the health care staff) to the patients, as well as to relatives and friends, in order that they become able to recognize the warning signs of suicidal behaviour, be aware of the risk periods and the importance of adherence to treatment, avoid isolation and call for help in emergency situations. A written list of sources of support which are available during a suicidal crisis may also be helpful [2,10,15,59].

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
