# Peer review of "Suicide Risk in Bipolar Disorder: A Brief Review"

_medicina, 2019, doi:10.3390/medicina55080403_

Round 1

Reviewer 1 Report

The authors have provided a brief review on suicide risk in bipolar disorder. The topic is of great interest and the manuscript provides a concise and clear overview.

Some aspects of the manuscript might benefit of changes. Here are some suggestions:

At page 1, line 42-43, please provide a reference for the following sentence: "Suicidal behaviour is quite frequent ... at least once in their lifetime".

At page 2, line 47-48, the following sentence is not clear and should be reformulated: "it is important to mention that the proportion of attemped and completed suicide is much lower ... ". Additionally, the sentence "one study, for example, reported that rate as 35:1 and 3:1 for the general population and for BD patients, respectively" needs a reference.

At page 2, line 63, the acronym MDD should not be explained as it has already been reported at line 53.

At page 2, line 79-93, page 3 lines 95-114, there is a very long paragraph lacking any reference. Are all the references for this paragraph reported in the last sentence at line 114? 

As this paragraph represents one of the most important parts of this review, it would be clearer for the reader if the authors provided references referring to the single factors they list to be associated with increased suicide prevalence next to the specific factors. Furthermore, in this paragraph the authors use sentences such as "these studies suggest that the majority of suicide risk..." and, without specific references for each sentence, it is not clear at all to which studies the authors are referring to.

Another aspect that needs to be addressed is the fact that this paragraph is merely a list of factors previously associated with suicide risk. It is not clear if all the listed factors are supported by the same level of evidence (e.g. have been shown to be associated with suicide risk by one or more studies, with similar sample size and so on) or if evidence for some of these factors is stronger/weaker/needs further replication. Also, it would be useful to report information on the effect sizes, if available, for these factors. 

At page 4, line 144, the expression "more superior" should be replaced with "superior"

These recent references might be included: 

10.1016/j.eurpsy.2019.06.001

10.1016/j.jpsychires.2019.03.001. 

Additionally, this relevant study should be cited and discussed: doi: 10.1111/bdi.12088.

Reviewer 2 Report

This short narrative review summarizes the literature on risk and treatment of suicide in people with bipolar disorder. Overall, the text is informative and written by experts in the field. Still, there are a few issues that need to be addressed. I list my concerns below.

Even a narrative review could use a ‘Method’ section. It would be helpful to know how the authors have searched, selected and analyzed the literature. In the current form, it looks like a random sample of the literature.

The risk and protective factors of suicide and attempted suicide in bipolar disorder appear to be similar to those reported in studies of the general population. Can the authors better differentiate the risk and protective factors that are more pronounced in bipolar vs the general population?

The heading ‘Summary’ at the end could be replaced with ‘Clinical implications’.

Overall, there is a major concern regarding language. In international literature, the verb ‘to commit’ is no longer used regarding suicidal behaviour. For example, line 18, ‘commit … suicide attempt’, could be ‘…attempt suicide’. See also line 35.

Similarly, the word ‘completed’ [suicide] could be replaced with ‘suicide death’.

‘Danger of suicide’ is ‘risk of suicide’.

The manuscript will benefit from proof-reading by a native English speaker to improve the flow of the sentences. For example, ‘Out of sociodemographic factors’ (lines 99-100) is not correct.

Line 31-32: ‘(see discussed below in details)’, can be deleted as this is the topic of your paper.

Good luck with the revision!

Round 2

Reviewer 2 Report

Thank you for submitting this revised version, and for considering my comments. This new version reads much better. I have only one question left.

Page 1, line 34: It says that 90% of suicides occur among patients with major mental disorders. Surely, this is an exaggeration. Most suicides are not even ‘patients’. It would be more correct to say that studies retrospectively have found that most people who had died by suicide had symptoms (or a diagnosis) of mental disorders. However, having symptoms does not equal meeting all criteria of a mental disorder. Please rephrase.

Otherwise there are no more questions. Good luck with the publication.
